# OpenReview forum: "Evaluating the Adversarial Robustness of Retrieval-Based In-Context Learning for Large Language Models"
_colmweb.org/COLM/2024/Conference — COLM_

### Official Review · Reviewer_YFEZ · 2024-04-18

**Rating:** 6
**Confidence:** 3
**Ethics Flag:** 1

**Summary:**

This paper focuses on evaluating the adversarial robustness of retrieval-based in-context learning (ICL) methods for large language models (LLMs). The authors conduct a comprehensive analysis of the vulnerability of different ICL methods to various types of adversarial attacks, including test-sample attacks, demonstration attacks, and datastore attacks. They also propose a training-free adversarial defense method called DARD, which enriches the example pool with adversarially attacked samples. The experimental results show that retrieval-based models can enhance robustness against test-sample attacks but are more vulnerable to demonstration attacks. DARD could improve the robustness of the models, achieving a reduction in attack success rate (ASR) over the baselines.

**Questions To Authors:**

Please see above.

**Reasons To Accept:**

* This paper is an early work studying the robustness of ICL methods, with interesting empirical findings.
* This paper provides a comprehensive analysis of the effects of perturbations on modern few-shot methods for LLMs inference. The experiments are generally sound and extensive.
* The proposed method is simple yet effective.
* This paper is well-written.

**Reasons To Reject:**

* This paper only studies the robustness of retrieval-based ICL in LLMs, and only focuses on the language modality. The scope of the topic may be limited. Moreover, does the robustness of LLMs significantly affect that of ICL? Why not study the robustness of LLMs instead? Is it possible that adversarially (and efficiently) fine-tuning LLMs (e.g., by LoRA) could yield better results? The authors should further highlight the motivation and significance of the studied topic.
* This is a pure empirical paper. It would be good to explore the theoretical part (e.g., certified robustness) in future work.
* No source code is provided at the time of submission.
* Minor: It seems that the font type is not correct.

---

> ### Author Rebuttal · Authors · 2024-05-29
>
> We appreciate the reviewer’s close reading of the paper and agree that our work provides certain insights. We respond to the reviewer’s concerns below.
>
> > This paper only studies the robustness of retrieval-based ICL in LLMs, and only focuses on the language modality. ...... Is it possible that adversarially (and efficiently) fine-tuning LLMs (e.g., by LoRA) could yield better results?
>
> We strongly believe that the need to study the robustness of ICL in LLMs is an important aspect of understanding the robustness of LLMs. In-context learning is still by far the most widely used technique for prompting the latest LLM methods [1]. It’s necessary to understand the robustness of different ICL methods and better study when ICL will degrade, given its input can be unintentionally or adversarially perturbed. We are unsure what the reviewer means by "robustness of LLM instead." We kindly ask the reviewer to elaborate slightly to clarify how studying ICL in LLMs does not classify as studying the robustness of LLMs. Furthermore, we include adversarial training with full parameters in Table 2 (the row with Adversarial Training) while proposing our method, DARD, to improve the robustness of ICL in LLMs.
>
>
> > This is a pure empirical paper. It would be good to explore the theoretical part (e.g., certified robustness) in future work.
>
> As the reviewer mentioned in the “Reasons to Accept,” our work is a pilot study on the robustness of ICL methods, and we hope to raise concerns among researchers in this field. We believe our major contribution is pointing out the problem of robustness in ICL for LLMs, supported by extensive experiments and insights. We believe our results will encourage further research on this topic, including providing theoretical support on why ICL fails and how it could potentially improve its robustness.
>
>
> > No source code is provided at the time of submission
>
> The code has been included as part of the supplementary materials at the time of submission. We will release all code and data in the preprint version to support the community.
>
>
> > It seems that the font type is not correct.
>
> That’s incorrect; we strictly followed the official template from COLM and used the original font type.
>
> [1] Agarwal, Rishabh, et al. "Many-Shot In-Context Learning." arXiv preprint arXiv:2404.11018 (2024).

---

> > ### Comment · Reviewer_YFEZ · 2024-06-04
> > **Thanks for your rebuttal**
> >
> > The authors solve my concerns, and I will keep my positive score.

---

> > > ### Author Response · Authors · 2024-06-04
> > > **Author response to Reviewer YFEZ**
> > >
> > > Dear Reviewer,
> > >
> > > Thank you very much for dedicating your time to review our response. If you found that our reply has satisfactorily addressed your concerns, we would be grateful if you could consider increasing your score. Your support would be greatly appreciated.
> > >
> > > Best regards,
> > >
> > > The Authors

---

### Official Review · Reviewer_xnUC · 2024-05-12

**Rating:** 7
**Confidence:** 5
**Ethics Flag:** 1

**Summary:**

The authors conduct a comprehensive evaluation of the robustness of different ICL approaches, i.e., vanilla, kNN-ICL, and retrieval-based ICL, against Test-Sample, Demonstration, and Datastore Adversarial Attacks. Their findings reveal that retrieval-based ICL enhances robustness against adversarial attacks on test samples; however, their robustness diminishes when demonstrations are perturbed. Additionally, the authors proposed DARD,  a novel training-free adversarial defense method aimed at improving the robustness of retrieval-based ICL against such attacks.

**Questions To Authors:**

1. What does ``involve the aggregation of results from retrievers" in the caption of Figure 3 mean?
2. Besides these word-level attacks, have you considered some sentence-level attacks, such as paraphrasing?

**Reasons To Accept:**

1. The authors conducted extensive experiments to evaluate the robustness of various ICL approaches against different adversarial attacks. Their research offers a significant and timely analysis of the robustness of contemporary ICL methods, particularly retrieval-based ICL. They provide detailed insights and an alternative perspective on recent studies of LLMs.
2. Additionally, the authors introduced DARD, a straightforward yet effective defense strategy proven to enhance the robustness of retrieval-based ICL against these adversarial attacks.
3. The paper is clearly written and well-organized. It is easy to follow the authors' ideas, i.e. Figure 1, and understand their experimental settings. The authors use clear figures and tables, e.g., Table 1&2 and Figure 2&3&4, to demonstrate their experimental results and findings. In the appendix, the authors include detailed descriptions of the experimental settings.
4. The authors have conducted an extensive literature review and thoroughly introduced the background information.

**Reasons To Reject:**

There are some minor issues in the current manuscript.
1. It would have been beneficial to include certain key concepts in the main text, such as the introduction of the attacks.
2. The title "Ablation Study" in section 4.3 does not appropriately reflect the concepts discussed.
3. The texts in Figure 1 are too small to read.
4. The authors should have included the statistics of the datasets they tested.

---

> ### Author Rebuttal · Authors · 2024-05-29
>
> We appreciate the reviewer’s close reading of the paper and the fact that the reviewer found our results insightful. We respond to the reviewer’s questions below.
>
> > It would have been beneficial to include certain key concepts in the main text, such as the introduction of the attacks.
>
> We agree with the reviewer that the main section's discussion is missing details about the introduction of the attacks or retrievers used, which we have included in Appendix A.3 and A.1, respectively. This is due to the submission's page limit, and we don’t have enough space to include them in the main context. If we are accepted and granted an extension page, we will ensure that they are included in the preprint version so that the reader can better follow the ideas.
>
> > The title "Ablation Study" in section 4.3 does not appropriately reflect the concepts discussed.
>
> We agree and plan to rename the section title to “Attack on Model Variants” to reflect the discussion better.
>
> > The texts in Figure 1 are too small to read.
>
> We apologise for that and will redraw the figures in the preprint version.
>
> > The authors should have included the statistics of the datasets they tested.
>
> The statistics of the datasets used are as follows and will be included in the preprint.
> | Datasets | No. of Labels | Train/Test Size | Avg. length |
> |----------|---------------|-----------------|-------------|
> | SST-2    | 2             | 67348/872       | 23.07       |
> | RTE      | 2             | 2489/277        | 96.03       |
> | MR       | 2             | 8530/1066       | 37.76       |
> | CR       | 2             | 3394/376        | 30.8        |
> | MNLI-mm  | 3             | 8832/982        | 55.9        |
> | TREC     | 6             | 5452/500        | 19.59       |
>
> > What does ``involve the aggregation of results from retrievers" in the caption of Figure 3 mean?
>
> Sorry for the confusion. The results shown in Figure 3 represent the mean attack success rate among the three retrievers we used (the standard deviation is displayed in the error bar).
>
> > Besides these word-level attacks, have you considered some sentence-level attacks, such as paraphrasing?
>
> Thank you very much for your suggestion. We will consider incorporating sentence-level attacks (StressTest[1] and CheckList[2]) in the next version.
>
> [1] Naik et al., Stress test evaluation for natural language inference. In ACL, August 2018.
>
> [2]  Ribeiro et al., Beyond accuracy: Behavioral testing of NLP models with CheckList. In ACL, July 2020.

---

> ### Author Response · Authors · 2024-06-05
> **Near the End of the Discussion Period**
>
> As the discussion period deadline approaches, we would like to ask whether our response addressed the concerns raised by the reviewer in the “Reasons to Reject” and question sections. We would be happy to provide any further clarifications.
>
> Once again, we thank the reviewers for their time in reviewing our work and for providing valuable suggestions.
>
> Best regards,
>
> Authors

---

### Official Review · Reviewer_uXax · 2024-05-14

**Rating:** 5
**Confidence:** 4
**Ethics Flag:** 1

**Summary:**

The authors introduce an effective training free adversarial defence method, DARD, which enriches the example pool with those attacked samples. It shows that DARD yields improvements in performance and robustness, achieving a 15% reduction in Attack Success Rate  over the baselines.

**Reasons To Accept:**

1. The authors perform a comprehensive and in-depth evaluation on variants of  In-Context Learning (ICL): vanilla, kNN-ICL, and retrieval-based ICL methods against Test-Sample, Demonstration and Datastore Adversarial Attacks.
2. Different attack methods are interesting.

**Reasons To Reject:**

1. The paper is more like of an empirical study of ICL methods under attacks. The novelty is limited. And the authors had better put the method section before experiment section.
2. The explored datasets are good. But it would be better to explore some real problem in the era of LLM, such as "Jailbreaking Black Box Large Language Models in Twenty Queries".

---

> ### Author Rebuttal · Authors · 2024-05-29
>
> We appreciate the reviewer’s close reading of the paper and that the reviewer found our results compelling. We respond to the reviewer’s questions below.
> > The paper is more like of an empirical study of ICL methods under attacks. The novelty is limited. And the authors had better put the method section before experiment section.
>
> The focus of our paper is on studying the robustness of large language models (LLMs) under attacks in retrieval-based in-context learning (ICL) scenarios. As far as we know, our work is among the early studies on the robustness of ICL attacks concerning LLMs, as reviewer YFEZ pointed out. Our paper's main focus is analyzing the robustness of ICL in LLMs. Therefore, we placed the analysis experiments at the beginning of the paper. Following this, we proposed a retrieval-based method called DARD to enhance the robustness of LLMs against adversarial attacks.
>
> > The explored datasets are good. But it would be better to explore some real problem in the era of LLM, such as "Jailbreaking Black Box Large Language Models in Twenty Queries".
>
> Jailbreak attacks are a meaningful direction in LLM safety that induce LLMs through interaction to violate established security policies or instructions and perform actions they shouldn't. However, our research takes a different approach by analyzing the sensitivity of large language models when the context for in-context learning or test examples is adversarially perturbed. This means that the input to the model may be syntactically incorrect (TB, TF, BA) or the demonstrations may be flawed (AdvICL, S-L, S-L(Fix)). This analysis helps us understand the behavior of LLMs under **out-of-distribution contexts**.
>
> In practical applications, this pertains to scenarios where users input incorrect words or cannot guarantee the 100% accuracy of the demonstrations. In contrast, jailbreak attacks are a form of red-teaming, aimed at bypassing the safety policies implemented during **the alignment stage** [1]. Our research and jailbreak attacks **address different scenarios**, each with its own practical significance.
>
> [1] Anil et al., Many-shot Jailbreaking, Anthropic, 2024

---

> ### Author Response · Authors · 2024-06-05
> **Near the End of the Discussion Period**
>
> As the discussion period deadline approaches, we would like to ask whether our response addressed the concerns raised by the reviewer in the “Reasons to Reject” and question sections. We would be happy to provide any further clarifications.
>
> Once again, we thank the reviewers for their time in reviewing our work and for providing valuable suggestions.
>
> Best regards,
>
> Authors

---

### Decision · Program_Chairs · 2024-07-10

**Decision:**

Accept

**Comment:**

The paper introduces DARD, a training-free adversarial defense method designed to enhance the robustness of retrieval-based in-context learning (ICL) methods for large language models (LLMs) against various adversarial attacks. The authors conduct a comprehensive evaluation of different ICL methods—vanilla, kNN-ICL, and retrieval-based ICL—against test-sample, demonstration, and datastore adversarial attacks. Their findings highlight the vulnerabilities and strengths of these methods, with DARD showing a significant improvement in reducing the Attack Success Rate by 15% over baseline methods.

The paper is well-written and well-organized, making it easy to follow the authors' arguments and understand their experimental setup. The inclusion of detailed figures and tables aids in the clear presentation of results. The comprehensive literature review and background information further strengthen the paper's foundation. Overall, the paper provides valuable empirical findings and a useful defense mechanism for enhancing the robustness of ICL methods in LLMs.

However, there are several areas where the paper could be improved. The scope of the study is restricted to language modality, and it would be beneficial to explore the robustness of LLMs more broadly. Additionally, key concepts and dataset statistics could be more prominently discussed in the main text rather than being relegated to appendices. The paper would also benefit from a clearer definition of the attacks and an improved explanation of the motivation and significance of studying ICL robustness specifically.